# Bayesian accounts of perceptual decisions in the nonclinical continuum of psychosis: Greater imprecision in both top-down and bottom-up processes

**Isabella Goodwin**[1]*, **Joshua Kugel**[2], **Robert Hester**[1], **Marta I. Garrido**[1,3]

**1** Melbourne School of Psychological Sciences, The University of Melbourne, Melbourne, Victoria, Australia, **2** School of Psychology and Psychiatry, Monash University, Melbourne, Victoria, Australia, **3** Graeme Clark Institute for Biomedical Engineering, The University of Melbourne, Victoria, Australia

* goodwin.isabella@gmail.com

**Data Availability Statement:** All data and analysis code written in support of this manscuript is availble at: https://osf.io/qh5ca/.

## Abstract

Neurocomputational accounts of psychosis propose mechanisms for how information is integrated into a predictive model of the world, in attempts to understand the occurrence of altered perceptual experiences. Conflicting Bayesian theories postulate aberrations in either top-down or bottom-up processing. The top-down theory predicts an overreliance on prior beliefs or expectations resulting in aberrant perceptual experiences, whereas the bottom-up theory predicts an overreliance on current sensory information, as aberrant salience is directed towards objectively uninformative stimuli. This study empirically adjudicates between these models. We use a perceptual decision-making task in a neurotypical population with varying degrees of psychotic-like experiences. Bayesian modelling was used to compute individuals' reliance on prior relative to sensory information. Across two datasets (discovery dataset n = 363; independent replication in validation dataset n = 782) we showed that psychotic-like experiences were associated with an overweighting of sensory information relative to prior expectations, which seem to be driven by decreased precision afforded to prior information. However, when prior information was more uncertain, participants with greater psychotic-like experiences encoded sensory information with greater noise. Greater psychotic-like experiences were associated with aberrant precision in the encoding both prior and likelihood information, which we suggest may be related to generally heightened perceptions of task instability. Our study lends empirical support to notions of both weaker bottom-up and weaker (rather than stronger) top-down perceptual processes, as well as aberrancies in belief updating that extend into the non-clinical continuum of psychosis.

## Author summary

Investigating psychotic-like experiences in non-clinical populations can aid our understanding of how and why altered perceptual experiences arise in psychosis. On one hand,

**Funding:** This project was funded by Melbourne School of Psychological Sciences internal grant scheme (MIG), with IG receiving the Australian Government Research Training Program PhD Scholarship. The funders had no role in the study design, data collection and analysis, decision to publish or preparation of the manuscript.

**Competing interests:** The authors have declared that no competing interests exist.

psychotic-like experiences might arise due to an overreliance on current sensory information in one's environment with a reduced regard for prior contextual information when making decisions or inferences about stimuli and events. Alternatively, this might be due to an overreliance on contextual information such as prior beliefs and expectations. In this study, these hypotheses are adjudicated in a perceptual decision-making task in which the uncertainty of prior beliefs and sensory information is manipulated. We found that people with greater psychotic-like experiences relied more on sensory information relative to prior expectations across the task. This was driven by the perception of greater uncertainty or unreliability associated with prior information. Psychotic-like experiences were also associated with the perception of greater uncertainty in sensory information, suggesting altered encoding of both prior and likelihood information. These findings show that alterations in belief updating extend into the non-clinical continuum of psychotic-like experiences, which provide important utility in understanding the occurance of psychosis.

## Introduction

Sensory processing under uncertainty is intrinsic to how we predict and engage with the environment to form coherent and accurate representations or beliefs about outcomes that facilitate behavioural updating. Neurocomputational accounts of schizophrenia propose mechanistic disruptions in information processing that lead to the formation of delusions and hallucinations [1,2]. Given that the world we live in is often clouded by noisy, ambiguous sensory inputs, the brain must create an internal representation or model that is used to infer the cause of this sensory information [3,4]. A predictive processing perspective assumes that perception is an inferential optimisation process whereby the cause of sensory events is established via a combination of both current sensory evidence (*likelihood*) and prior beliefs or expectations (*priors*) about the occurrence of the event [3,5,6]. As incoming information may differ in its reliability across contexts, it must be weighted accordingly to facilitate accurate perceptual inference. Discrepancies between predictions and observations create error signals (prediction errors) that guide the updating of predictions and the relative precision attributed to expectations, compared to sensory information [3]. Aberrancies in this precision afforded priors and likelihoods provide a succinct mechanistic platform for understanding dysfunctions in perceptual inferences and experiences across the continuum of psychosis [2,7,8].

Early accounts of predictive processing suggest that an overreliance on sensory information relative to prior expectations confounds perceptual experiences in schizophrenia [1,9]. In this view, heightened salience placed on objectively uninformative events or stimuli results in the misinterpretation of prediction errors as meaningful change, which contributes to misleading belief updating about the environment. This can be explained as a failure to attenuate the precision of sensory information relative to prior beliefs at low hierarchical levels, leading to perception being overly driven by bottom-up processes [2]. Contrasting accounts suggest that aberrant perceptual experiences arise due to an over-reliance on prior expectations or beliefs about the cause of sensory events, driven by an abnormality in top-down guided perception [10,11]. This results in an over-attenuation of prediction errors, with overly precise priors contributing to perception in the absence of objectively identifiable stimuli. Extensive evidence has provided support for both sides of these seemingly contradictory canonical accounts of hallucination and delusion formation. More recently, the hierarchical nature of predictive coding processes have been utilised to harmonise these competing hypotheses into a unifying explanatory framework [2,12–14].

Although these co-occurring aberrancies can be accounted for within the hierarchical nature of predictive coding, it is somewhat unclear how the precision afforded to each type of information is altered. For example, an overreliance on sensory information could be driven by chronically over-precise low-level prediction errors contributing to unusual belief updating [15]. This would mean that more weight or salience is placed on persistently surprising events, requiring the adoption of higher-order beliefs for them to be explained away [16]. Alternatively, higher-order beliefs may be of low precision, leading to a lack of regularisation that renders the environment seemingly volatile or unpredictable, thus enhancing the weight of lower-level prediction errors [1,2]. Thus, discrepancies within the literature describe alternative accounts of how precision is afforded to likelihood and prior information. It is unclear whether these differences are driven by aberrant precision of encoding uncertainty in likelihood or prior information. Our study is designed to empirically adjudicate between these opposing accounts. We do so by directly quantifying the precision encoded in both prior and likelihood information in neurotypical individuals with varying levels of psychotic-like experiences.

Recognising the complexity and dimensional nature of psychopathology, contemporary approaches into the investigation of psychoses signify a shift towards dissecting the spectrum of schizophrenia into subgroups or dimensions and even beyond dichotomous notions of health and disease [17]. This view suggests that perceptions about instability of the world may gradually extend into non-clinical populations who have psychotic-like experiences, rather than presenting at a threshold level for diagnostic classification. Within this psychosis-continuum perspective, psychotic symptoms are considered to be an extreme outcome of a continuously distributed phenotype which extends into the non-clinical population [18]. Thus, understanding aberrant information integration along the continuum of psychosis can elucidate the development of proneness to hallucinations and delusions, into more extreme symptomatologic presentations in schizophrenia.

Here, we used Bayesian modelling to quantify the precision of representations of likelihood and prior information in a task that manipulated uncertainty in both [19], providing a deeper elucidation of perceptual inference which is lacking in previous literature [20]. This allowed us to disentangle between the competing theories of the development of psychotic-like symptoms, namely a top-down, overreliance on priors account, and a bottom-up, overreliance on likelihood account. Our approach enabled us to determine whether an overreliance on prior or likelihood information correlates with psychotic-like experiences. It also allowed us to elucidate whether the subjective uncertainty associated with the representation of likelihood and/or prior information (a proxy for the precision afforded to each source of information) was associated with psychotic-like experiences. Additionally, investigating variability in the relative weighting of prior to likelihood information across the task allowed us to determine the relationship between psychotic-like experiences and aberrancies in the stability of information integration in this paradigm. In response to the replication crisis, we present parallel results from a discovery and an independent, validation dataset.

## Methods

### Ethics statement

All participants gave written informed consent and received monetary compensation (5GBP) for participation. The study was approved by the University of Melbourne Human Research Ethics Committee (Ethics ID: 20592).

## Participants

The total sample from the discovery dataset included 363 participants (age range 21–80, M = 44.02, SD = 13.31; 150 male, 210 female, 3 non-binary/prefer not to say). A power analysis was conducted based on the findings of the discovery data (R software: "pwr.r.test" function), indicating that, to increase the power to 80% to observe small to moderate effects at $a = 0.05$ two-sided, an optimal sample size of 782 participants would be required for the validation dataset. Accordingly, the total sample from the validation study included 782 participants (age range 18–73, M = 31.62, SD = 11.15, 356 female, 412 male, 14 non-binary/prefer not to say).

All participants were recruited through Prolific (www.prolific.co), an online platform widely used to source participants in the general population. To be eligible for the study, participants had to be at least 18 years old and have corrected to normal vision. Participants were asked about their highest level of education, left or right handedness, previous diagnosis of neurological conditions, emotional or psychological disorders, and/or substance dependence, previous drug use, and any other conditions that may affect performance.

## Experimental design

### Procedure

Firstly, participants provided demographic details and completed the 42-item Community Assessment of Psychic Experiences (CAPE; [21]), used to measure subclinical psychotic-like experiences via Qualtrics (www.qualtrics.com). The CAPE uses a four-point Likert scale to measure lifetime psychotic experiences. It measures frequency and distress of psychotic-like thoughts, feelings, and mental experiences. Specifically, it relates to three subdimensions of psychosis symptomatology, assessing alterations in thought and sensory perception (positive dimension), social isolation and affective blunting (negative dimension), and anhedonia and sadness (depressive dimension). Since we were interested in drawing parallels between the accounts proposed for positive symptoms in schizophrenia and those in nonclinical individuals, this study focused on the positive symptom frequency subscale (CAPE-P) as our primary measure of psychotic-like experiences. Following completion of this questionnaire, participants were then directed to Pavlovia (www.pavlovia.org), where they completed a computerised spatial span test [22] as a measure of working memory. Finally, participants completed the perceptual decision-making tasks, involving the likelihood-only estimation task followed by the coin task as modified from Vilares et al., [19].

### Coin task

Participants performed a decision-making task where they were asked to guess the position of a hidden target on a screen, requiring them to integrate both noisy sensory evidence of the target's location, and prior expectation of the target's location. More specifically, participants were told a coin was being thrown into a pond and were asked to guess where the coin had fallen. Likelihood and prior variance were manipulated with a two-by-two factorial design with narrow and wide variance respectively. On each trial, five blue dots denoted "splashes" produced by the coin falling in. The variance of these splashes changed on each trial as an index of either narrow or wide likelihood conditions. The position of these splashes was drawn from a Gaussian distribution centred on the (hidden) location of the coin, with standard deviation of either 6% of the screen width (SD = 0.096; narrow likelihood trials) or 15% of the screen width (SD = 0.24; wide likelihood trials). An example trial is shown in Fig 1. Participants were also informed that the person throwing the coin changed between blocks, and one thrower was more accurate than the other. They were told that both throwers aimed at the screen

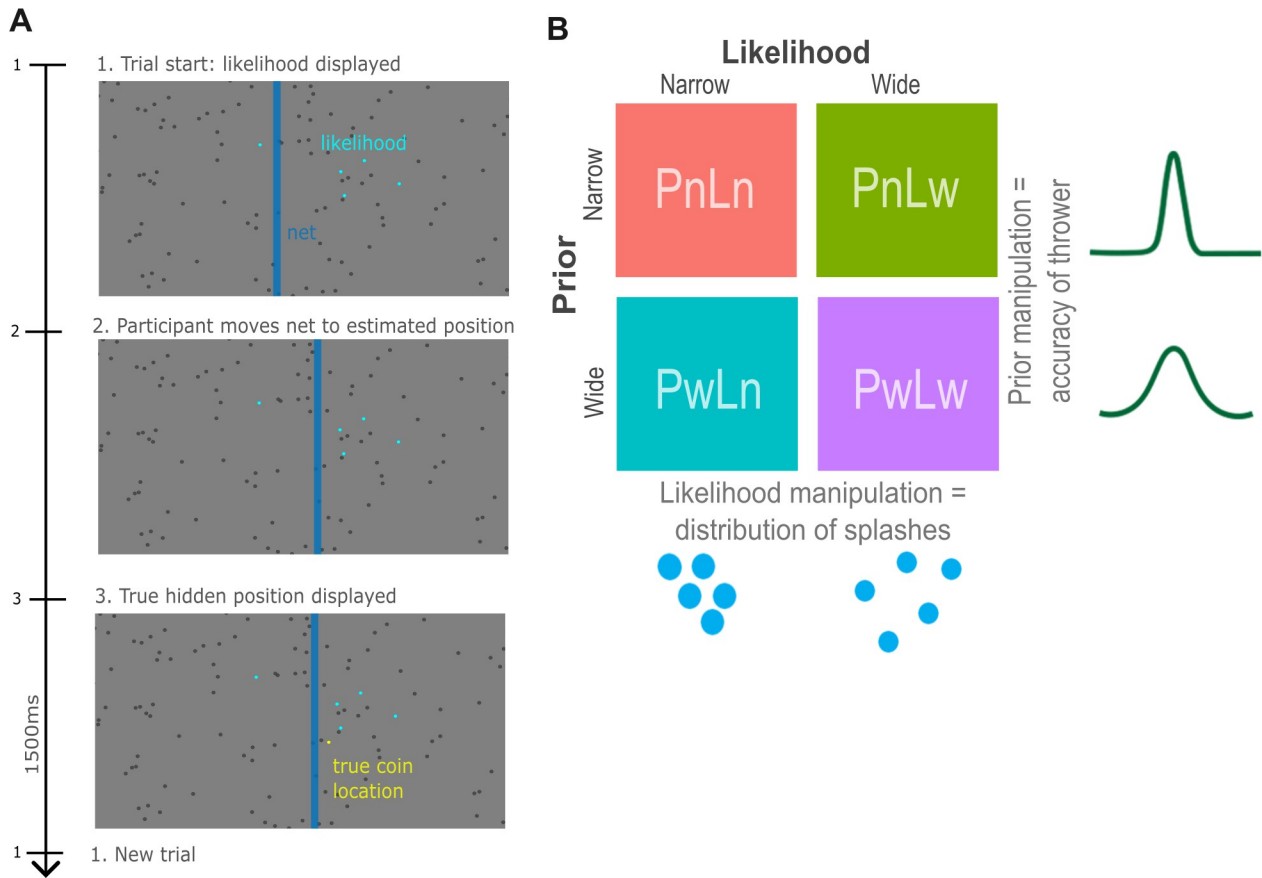

**Fig 1. Coin task paradigm (adapted from Vilares et al., [19] and Randeniya et al., [23]).** *A)* Time course of a single exemplar trial: participants are shown five blue dots to represent splashes of the location of a coin being thrown into a pond. They are then asked to move the blue bar/net to where they estimate the coin's location to be, after which the coin's true location is revealed, and they move onto the next trial. *B)* Task design as adapted from Vilares et al., [19]: the four conditions of the task are visually depicted including two types of likelihood as manipulated through the distribution of splashes on each trial (Ln = narrow likelihood; Lw = wide likelihood) and two types of prior as manipulated through the accuracy of the thrower on each block (Pn = narrow prior; Pw = wide prior).

centre (indicating the mean of the prior). Although they were not explicitly told which thrower was better or worse, this could be inferred through the distribution of previous coin locations from trial-to-trial. Although participants were not explicitly told which block they were in, they were informed that thrower A and thrower B alternated between blocks. This was randomised across participants so that half of participants received thrower A first, and half the participants received thrower B first. Thus, participants could infer the prior information (regarding the variability of the thrower) based on the distribution of throws. The location of the coin was drawn from a second, independent Gaussian distribution centred on the middle of the screen, with a standard deviation of either 2.5% of the screen width ($SD = 0.04$; narrow prior blocks) or 8.5% of the screen width ($SD = 0.136$; wide prior blocks). The four conditions are visually depicted in Fig 1.

While the variance of the likelihood changed pseudorandomly from trial-to-trial (counterbalanced across all trials), the variance of the prior changed from block to block, with the order (thrower A vs thrower B) also counterbalanced across participants. Thus, there were four conditions: narrow prior and narrow likelihood (PnLn), narrow prior and wide likelihood (PnLw), wide prior and narrow likelihood (PwLn), and wide prior and wide likelihood (PwLw).

For each trial, participants were instructed to move a net (blue bar) horizontally across the screen to indicate where they thought the coin had landed. The true position of the coin (represented as a yellow dot) was then shown for 1500 ms. Scoring was tallied across each trial, with a point earned each time any part of the coin lay within the net. Participants were provided with two blocks of two practice trials before completing the main task. The main task consisted of two blocks per thrower, with each block containing 75 trials each (resulting in 300 trials total).

## Likelihood only task

Prior to completing the coin task, participants completed the likelihood-only estimation task to determine a measure of subjective likelihood variance or sensory noise. The setup of this task was identical to the main task, without the incorporation of the prior condition. This provided an estimation of how participants perceived the centre of the splashes on their own, without prior knowledge. Participants were asked to estimate where they thought the true coin location was, which was always the centre of the displayed splashes, by moving the net horizontally across the screen. The true coin location (represented as a yellow dot) was revealed at the end of each trial, providing feedback on participants estimations. This task consisted of 100 trials, with an even number of wide and narrow likelihood distributions.

## Behavioural analysis

Successful performance of the task required participants to move the net to the most likely location of the hidden coin. Utilising Bayes rule, we can elucidate what the optimal estimate of the position of the coin would be on each trial [19,24]:

$$X_{est} = \frac{\sigma_L^2}{\sigma_L^2 + \sigma_P^2}\mu_P + \frac{\sigma_P^2}{\sigma_L^2 + \sigma_P^2}\mu_L \qquad (1)$$

where $X_{est}$ is the estimated position of the coin (i.e., participants responses on each trial), ($\mu_P$, $\mu_L$) represent the prior and likelihood means and ($\sigma_P^2$, $\sigma_L^2$) represent the prior and likelihood variances, respectively. In our experiment, the mean of the prior was kept constant (the centre of the screen, $\mu_P$), while the mean of the likelihood was determined by the centre of the five blue dots in each trial ($\mu_L$).

## Performance

Performance in the likelihood-only task was characterised by the average distance between participants estimates of the coin location (net location) and the true centre of the splashes (i.e., mean estimation error). Similarly, performance in the coin task was characterised by the average distance participants estimates (net location) and the true location of the coin.

## Sensory weight (likelihood vs prior reliance)

To estimate participants reliance on likelihood relative to prior information, we fitted a linear regression to participants' estimates of the coin's position for each trial ($X_{est}$) as a function of the centre of the splashes (i.e., the likelihood mean, $\mu_L$):

$$sw = \frac{\sigma_P^2}{\sigma_L^2 + \sigma_P^2} \qquad (2)$$

where $sw$ is the slope of the linear regression, which indicates how much each participant relies on likelihood information. A slope closer to one indicates a greater reliance on the likelihood

information, while a slope closer to zero indicates greater reliance on prior information. A slope between zero and one indicates that participants integrate both likelihood and prior information in their estimates. This was calculated overall, for each condition, and for each block. We use the term sensory weight throughout the manuscript. Note, however, that the sensory weight is a relative measure for how much people rely on sensory information versus prior information, hence it is simultaneously a measure of "*reliance on sensory information*" and "*reliance on prior information*".

## Bayesian optimal sensory weights

If participants perform according to the Bayesian optimum as portrayed in Eq (1), then the optimal values for the slopes/sensory weights should be equal to the perceived $\frac{\sigma_P^2}{\sigma_L^2 + \sigma_P^2}$, where $\sigma_P^2$ is the variance associated with the prior (narrow prior $\sigma_P^2 = 0.04^2$; wide prior $\sigma_P^2 = 0.136^2$) and $\sigma_L^2$ is the variance associated with the likelihood (in this instance, narrow likelihood $\sigma_L^2 = \frac{0.096^2}{5}$; wide likelihood $\sigma_L^2 = \frac{0.24^2}{5}$). These calculations of Bayesian optimality refer to posterior computations, integrating the relative uncertainty of both prior and likelihood information.

## Sensitivity to prior change across blocks

An examination of how sensitive participants were to changing prior uncertainties can be estimated by the mean absolute difference in sensory weight from one block to the next ($B_i = 1,\ldots,4$):

$$Sensitivity\ to\ prior\ change = \frac{|sw_{B1} - sw_{B2}| + |sw_{B2} - sw_{B3}| + |sw_{B3} - sw_{B4}|}{3} \qquad (3)$$

which provides an indication of individuals' sensitivity to changes in prior uncertainty across the task.

## Trial-by-trial variability in sensory weight

Eq (1) can be rewritten to calculate an instantaneous sensory weight as an indicator of participants reliance on likelihood relative to prior information on any given trial:

$$sw_{trial} = \frac{X_{est} - \mu_P}{\mu_L - \mu_P} \qquad (4)$$

where $X_{est}$ is the participants estimated position of the coin on a given trial (net location), $\mu_P$ is the mean of the mean of the prior (assumed at the centre of the screen), and $\mu_L$ is the mean of the likelihood (the centre of the five blue dots for that trial). To ensure the trial-by-trial sensory weight varied from 0 to 1, we calculated the logistic of the $sw_{trial} = \frac{1}{(1+e^{-swtrial})}$. This variance of the trial-by-trial sensory weight for each participant over the whole session provided an indication of how much participants sensory weights varied instantaneously across the task.

## Subjective likelihood variance

The likelihood-only task can be used to determine a proxy for participants subjective likelihood variance or sensory noise [23]. This is determined by the variance of the participants

estimates of the mean ($\mu_{est}$) relative to the true mean of the splashes ($\mu_L$):

$$\sigma_{SL}^2 = \frac{\Sigma(\mu_{est} - \mu_L)^2}{nTrials} \tag{5}$$

where the number of trials (nTrials) was equal to 100 in the likelihood-only task.

## Subjective prior variance

Subjective prior variance ($\sigma_P^2$) is an estimate of participants' subjective representation of the variability in prior information (i.e., the variability of the thrower). This is expected to change between blocks depending on which thrower is throwing the coin (as one is more variable than the other) and can be used as a proxy to determine participants' individual estimates of how much variability or overall uncertainty there is in prior information. Subjective prior variance is calculated as a weighted combination of subjective likelihood variance and sensory weight, which is derived by rearranging Eq 2 as follows:

$$\sigma_P^2 = \frac{\sigma_{SL}^2 * sw}{(1 - sw)} \tag{6}$$

Higher values for the subjective prior variance should be interpreted as more subjective uncertainty (i.e., increased variability) in the internal representations of prior information. In this equation, $\sigma_{SL}^2$ is estimated from participants subjective likelihood variance (as calculated in Eq 5). Optimal or 'imposed' prior variance scores were also calculated based on this equation, where the actual likelihood variance and optimal sensory weights replaced the perceived values (see section on Bayesian optimal sensory weights for calculations).

## Statistical analysis

To understand whether Bayesian models can explain sensory learning in psychotic-like experiences, we explored the relationship between CAPE-P scores and aspects of task performance. Spearman-ranked correlation analyses were calculated for non-parametric variables, whilst Pearson correlations were calculated for linearly distributed variables (such as log-transformed CAPE-P scores). Additionally, bootstrapped confidence intervals with 1000 bootstrapped replicates were calculated for these correlation analyses. Although we believe that accounting for multiple comparisons with Bonferroni corrections is too conservative of an approach for our research design and may increase the risk of a type 2 error [25], we have included adjusted p-values for our main correlations of interest as a comparison to bootstrapped confidence intervals in S3 Table. Mean estimation error in the likelihood only task was used as a criterion to detect poor performance or low effort, with 7 outliers excluded from the discovery dataset and 12 outliers excluded from the validation dataset (z score greater than ± 3). Similarly, mean estimation error in the main task was used to detect poor performance, with 8 outliers excluded from the discovery dataset and 27 outliers excluded in the validation dataset.

## Results

### Participants

Data from a total of 1145 participants were collected across the two datasets, with demographic details provided in Table 1. Interestingly, there was a significant difference between CAPE scores (total scores as well as each subscale scores) across the two datasets (provided in Table 1). Specifically, the mean CAPE-P score was significantly higher in the validation dataset compared to the discovery dataset (Fig 2). Despite this, a previously published mean CAPE-P

**Table 1. Demographic profiles and scores from the CAPE across the discovery and validation datasets, with p-values indicating differences between the two datasets.**

| | Discovery dataset (n = 363) | | | Validation dataset (n = 782) | | | |
|---|---|---|---|---|---|---|---|
| | **M** | **SD** | **Range** | **M** | **SD** | **Range** | **p-value** |
| Age (years) | 44.02 | 13.31 | 21–80 | 31.62 | 11.15 | 18–73 | |
| Gender | Female | Male | Other | Female | Male | Other | |
| | 57.9% | 41.3% | 0.8% | 45.5% | 52.7% | 1.8% | |
| Highest level of education | Primary school | Secondary school | Tertiary education | Primary school | Secondary school | Tertiary education | |
| | 1.1% | 22.3% | 76.5% | 0.9% | 26.3% | 72.5% | |
| Employment status | Full time | Part time | Unemployed | Full time | Part time | Unemployed | |
| | 49.3% | 18.7% | 10.7% | 41.4% | 16.7% | 17.2% | |
| CAPE Total | 63.71 | 14.59 | 42–120 | 73.06 | 15.66 | 42–130 | $2.2 \times 10^{-16}$ |
| CAPE-P | 25.07 | 5.38 | 20–55 | 30.32 | 7.61 | 19–63 | $2.2 \times 10^{-16}$ |
| CAPE-N | 24.51 | 7.16 | 14–46 | 27.14 | 7.16 | 14–52 | $2.26 \times 10^{-09}$ |
| CAPE-D | 14.13 | 4.21 | 8–31 | 15.61 | 4.36 | 7–31 | $6.32 \times 10^{-10}$ |

*Note*: CAPE-P = positive dimension, CAPE-N = negative dimension, CAPE-D = depressive dimension

score in a large sample (N = 21,590) with online administration ($M = 27.7$, $SD = 4.5$; [26]) lay between the mean scores from the discovery ($M = 25.07$, $SD = 5.38$) and validation ($M = 30.32$, $SD = 7.61$) datasets. The country of residence of responders differed slightly across the two datasets, with the majority in the discovery dataset residing in the United Kingdom (75.7%) and the United States (17.4%), compared to the majority from the validation dataset residing in South Africa (22.4%) and Mexico (19.7%), followed by the United Kingdom (13.17%) and the United States (8.06%). Interestingly, psychotic-like experiences as measured by the CAPE have previously been found to be more frequent in low and middle income countries than in high income countries [27]. This could account for the differences in CAPE-P distributions across the two samples, potentially increasing the generalisability of our findings.

## Task performance

An analysis of performance accuracy in the likelihood-only task revealed significantly greater estimation errors in the wide likelihood condition, compared to the narrow likelihood

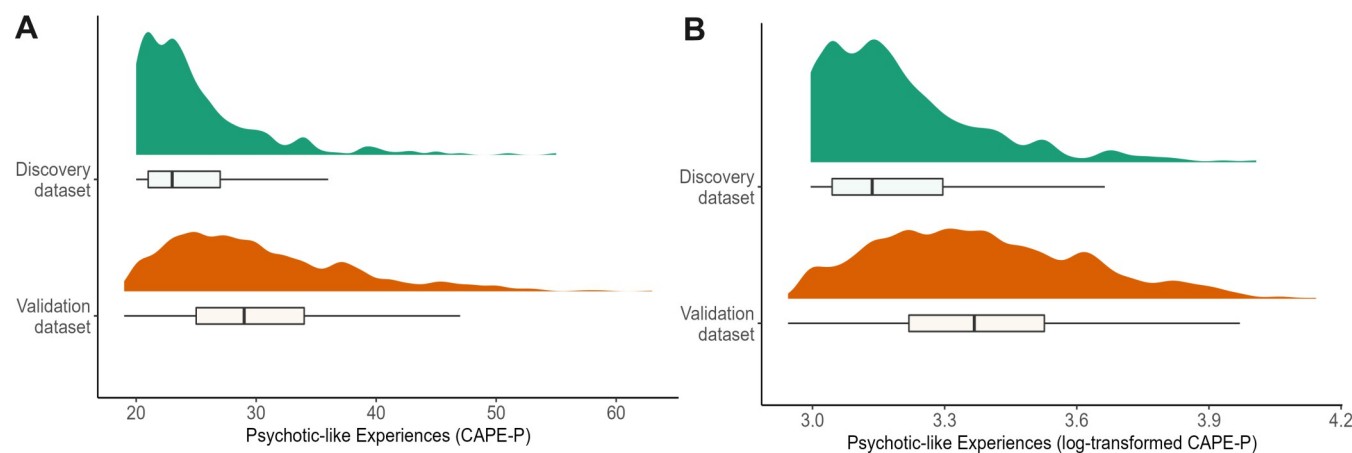

**Fig 2. Distribution of psychotic-like experiences.** Measured via CAPE-P scores in both discovery and validation datasets, including A) raw scores and B) log-transformed scores.

condition for both the discovery ($t(355) = 27.24$, $p < 2.2$ x $10^{-16}$, 95%CI = [0.016, 0.019) and validation ($t(769) = 43.09$, $p < 2.2$ x $10^{-16}$, 95%CI = [0.017, 0.019]) datasets. This indicated that the likelihood uncertainty manipulation functioned as expected, with higher uncertainty in the wide likelihood condition resulting in greater estimation errors. Additionally, a two-way ANOVA of task performance in the main task revealed no significant differences in mean estimation errors in the wide vs narrow prior condition ($F = 1.564$, $p = 0.211$), and the wide vs narrow likelihood condition ($F = 0.014$, $p = 0.906$), in the discovery dataset. In the validation dataset, a main effect of prior (Pw>Pn; $F = 6.34$, $p = 0.011$), indicating that greater estimation errors occurred in the wide prior condition compared to the narrow prior condition, however there was no significant difference in mean estimation errors occurring in the wide likelihood and narrow likelihood conditions ($F = 2.489$, $p = 0.114$; see S1 Fig for comparison across conditions).

### Participants rely on the most reliable source of information aligning with Bayesian principles, albeit non-optimally

A comparison of mean sensory weights between conditions was used to establish relative weighting on likelihood to prior information in the main task. A two-way ANOVA revealed a main effect of prior (Pw>Pn; discovery: $F = 282.96$, $p < 2.2$ x $10^{-16}$; validation: $F = 531.31$, $p < 2.2$ x $10^{-16}$), with higher sensory weights in the wide prior condition indicating that participants were more likely to rely on likelihood information when the prior was more variable. Similarly, a main effect of likelihood (Ln>Lw; discovery: $F = 81.76$, $p < 2.2$ x $10^{-16}$; validation: $F = 126.68$, $p < 2.2$ x $10^{-16}$) indicated that participants were more likely to rely more on likelihood information when the likelihood was less variable. This was replicated across the two datasets, indicating that reliance on likelihood relative to prior information functioned as expected across the four conditions (Fig 3). Moreover, Wilcoxon signed rank tests indicated that participants median likelihood to prior weightings significantly differed from Bayesian optimal weightings (blue dashed lines in Fig 3) in each condition and across both datasets (see S1 Table). Despite this, the pattern of performance was verging towards optimal, suggesting that participants were approximating Bayesian performance.

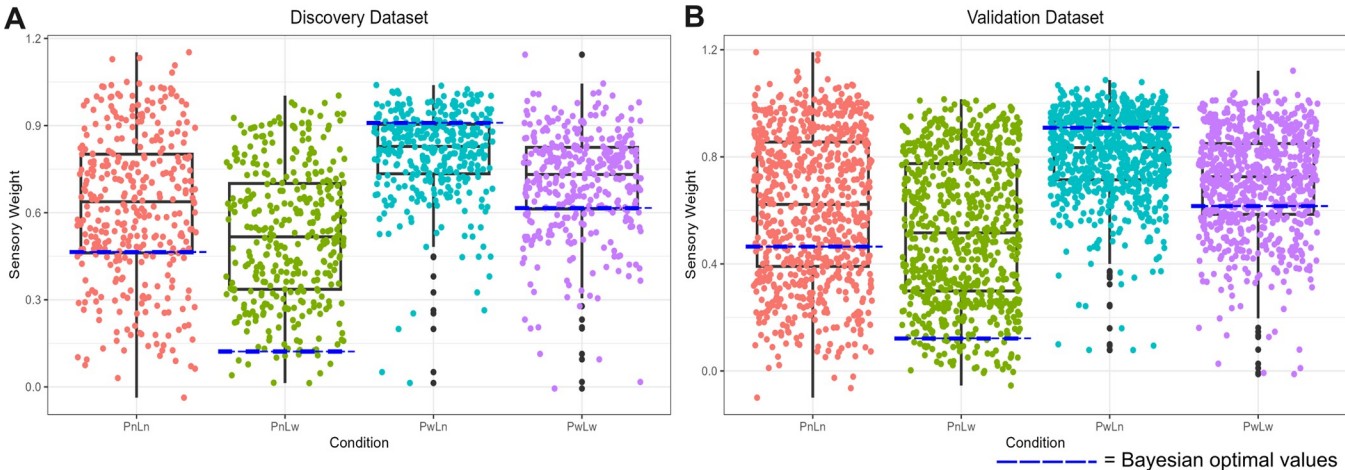

**Fig 3. Comparison of sensory weights across the four conditions.** Sensory weight for each participant is calculated by the slope of the regression between the true centre of the likelihood and participant's estimates of the coin's location for each condition. Sensory weight closer to 1 indicates greater reliance on likelihood, whilst sensory weight closer to 0 indicates greater reliance on prior. Blue dashed lines indicate Bayesian optimal computations of the coin's location, based on the posterior integration of uncertainty in both prior and likelihood information. Replicated patterns of performance were found across the A) discovery and B) validation datasets relative to Bayesian optimality. Conditions: PnLn = narrow prior, narrow likelihood (red dots); PnLw = narrow prior, wide likelihood (green dots); PwLn = wide prior, narrow likelihood (teal dots); PwLw = wide prior, wide likelihood (purple dots).

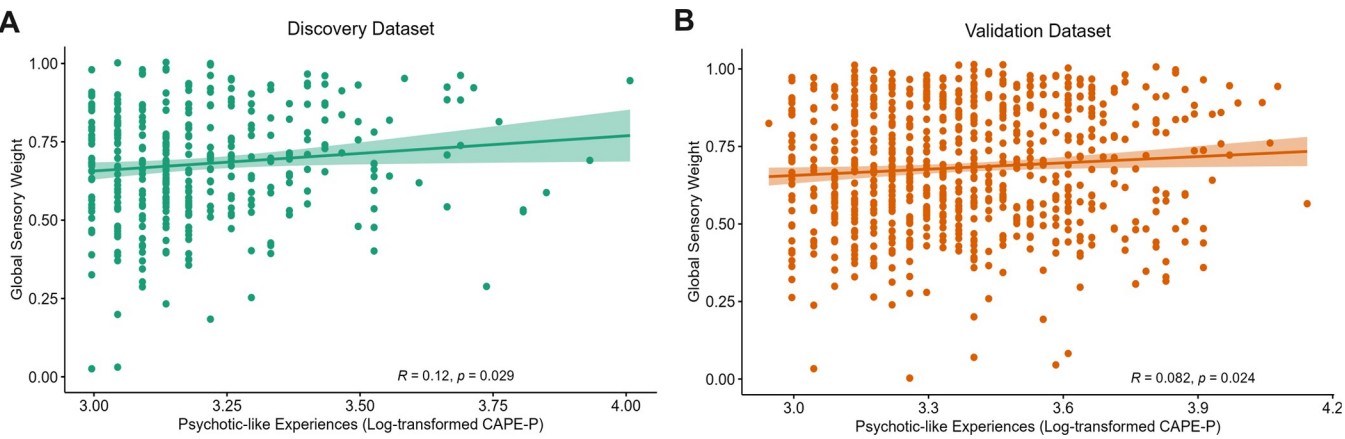

**Fig 4. Association between psychotic-like experiences and global sensory weight.** A significant positive relationship was found between psychotic-like experiences and global reliance on likelihood relative to prior information across the task in both the A) discovery and B) validation datasets.

## Psychotic-like experiences positively associated with increased reliance on likelihood information

Pearson correlation between log-transformed CAPE-P scores and global sensory weights (i.e., reliance on likelihood to prior information across the task) revealed a significant, positive relationship in both the discovery ($r = 0.12$, $p = 0.029$, 95%CI = [0.015, 0.221]) and validation ($r = 0.082$, $p = 0.024$, 95%CI = [0.0052, 0.157]) datasets (Fig 4). This indicates that psychotic-like experiences are associated with an increased reliance on likelihood information across the task. When considering the relationship between log-transformed CAPE-P scores and sensory weights in each condition, a significant positive relationship was found in the narrow prior conditions, but not the wide prior conditions across both datasets (provided in Table 2).

## No significant relationship between psychotic-like experiences and variability in likelihood vs prior reliance across blocks and from trial-to-trial

Average change in sensory weight was computed across blocks, providing an indication of individuals' sensitivity to changes in prior uncertainty (which changed every block). An analysis of average slope change across blocks revealed a weak, negative Spearman correlation between sensitivity to prior change and CAPE-P scores in the validation dataset ($r = -0.074$, $p = 0.042$), but no significant relationship in the discovery dataset ($r = -0.056$, $p = 0.29$). Similarly, instantaneous changes to sensory weights provided an indication of how much one's reliance on likelihood to prior information varied from trial-to-trial. No significant relationship

**Table 2. Pearson correlation analyses between sensory weights (relative likelihood to prior reliance) and log-transformed CAPE-P scores across each of the four conditions.**

| | Discovery dataset | | Validation dataset | |
|---|---|---|---|---|
| *Transformed CAPE-P score correlated with*: | $r_s$ | $p$ | $r_s$ | $p$ |
| | Sensory weight | | Sensory weight | |
| **PnLn** | 0.112 | 0.034 | 0.075 | 0.038 |
| **PnLw** | 0.109 | 0.041 | 0.083 | 0.022 |
| **PwLn** | 0.111 | 0.037 | 0.058 | 0.112 |
| **PwLw** | 0.069 | 0.192 | 0.021 | 0.567 |

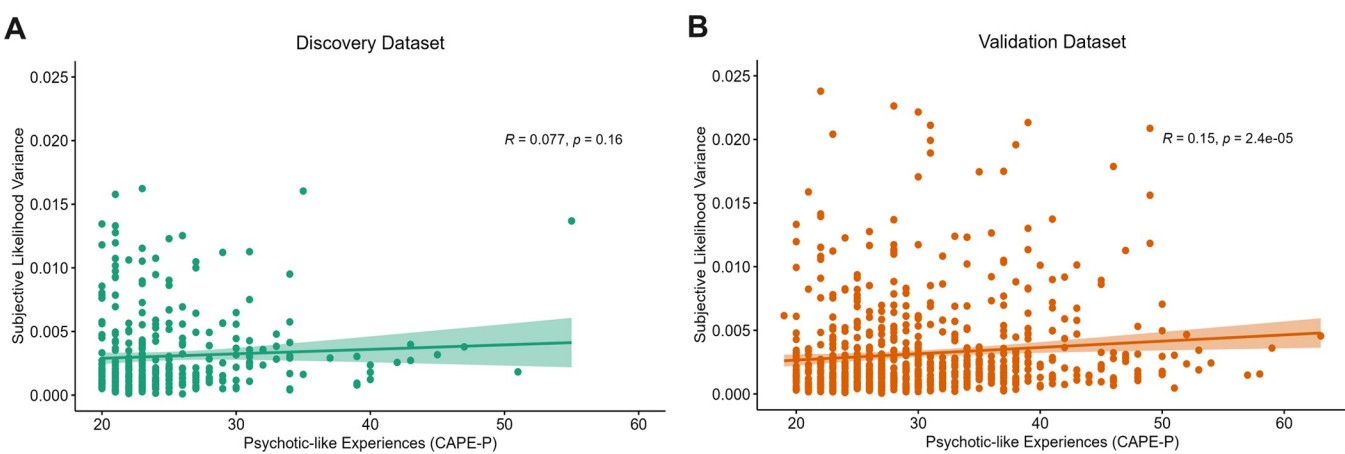

**Fig 5. Association between psychotic-like experiences and subjective likelihood variance.** A positive relationship was found between subjective likelihood variance across the task and increasing psychotic-like experiences in B) the validation dataset, but not A) the discovery dataset.

was found between variance in trial-by-trial sensory weight across the task and CAPE-P scores for both the discovery ($r = 0.096$, $p = 0.069$) and validation ($r = 0.064$, $p = 0.079$) datasets. This suggests there was little to no relationship between psychotic-like experiences and variability or changes to sensory weights across blocks, and instantaneously across trials.

## Psychotic-like experiences positively associated with subjective likelihood variance in validation dataset only

Subjective likelihood variance was calculated in the likelihood-only task as a proxy of how much participants perceived uncertainty in likelihood information (i.e., distribution of the five blue dots) to varying across the task. As expected, participants' average subjective likelihood variance was found to be significantly greater in the wide likelihood condition, compared to the narrow likelihood condition in both the discovery ($t(339) = 11.06$, $p < 2.2$ x $10^{-16}$) and validation ($t(750) = 12.81$, $p < 2.2$ x $10^{-16}$) datasets. When considering the relationship between overall subjective likelihood variance (across the likelihood-only task) and psychotic-like experiences, no significant Spearman-ranked correlation was found in the discovery dataset ($r = 0.077$, $p = 0.16$, 95%CI = [-0.031, 0.182]; Fig 5), whilst a significant positive Spearman-ranked correlation between subjective likelihood variance and CAPE-P scores was found in the validation dataset ($r = 0.15$, $p = 2.5$x$10^5$, 95%CI = [0.084, 0.224]). This might suggest that participants with increasing psychotic-like experiences are more likely to perceive the likelihood information to be more uncertain or variable across the task, however this effect was only found in the larger dataset.

## Psychotic-like experiences positively associated with subjective prior variance across both datasets

Furthermore, individuals' subjective likelihood variance and sensory weights were utilised to calculate subjective prior variance, as a proxy for how much participants were perceiving uncertainty in prior information (the accuracy of the thrower) to vary across the task. A comparison of mean subjective prior variance between conditions revealed a main effect of prior (Pw>Pn; discovery: $F = 46.95$, $p = 1.18$ x $10^{-11}$; validation: $F = 47.94$, $p = 5.56$ x $10^{-12}$), suggesting that participants were more likely to estimate that prior information was more variable in the wide prior condition, compared to the narrow prior condition. Despite this expected

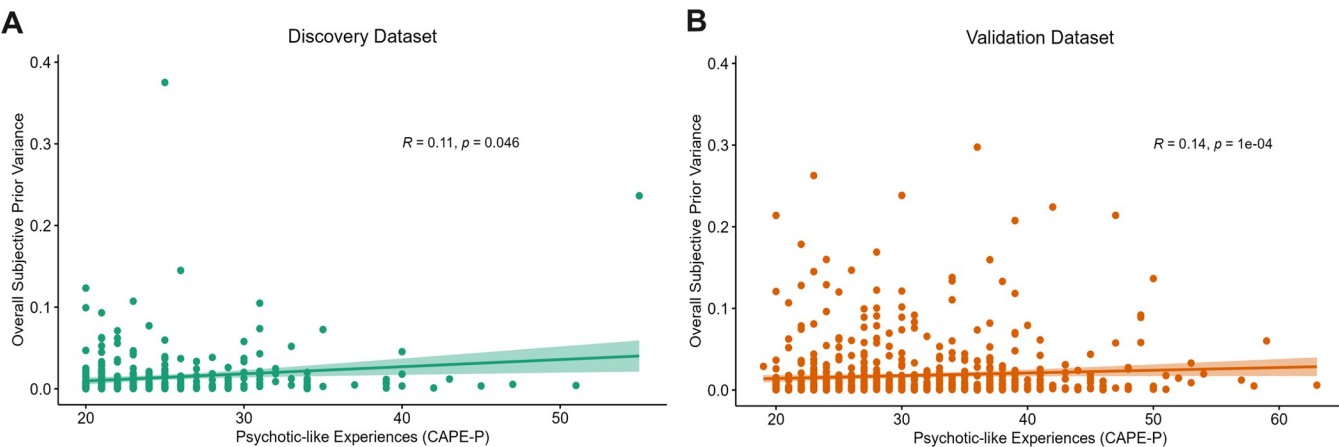

**Fig 6. Association between psychotic-like experiences and subjective prior variance.** A significant positive relationship was found overall subjective prior variance across the task and increasing psychotic-like experiences in both the A) discovery and B) validation datasets.

pattern of performance, Wilcoxon ranked test indicated that participants' median subjective prior variance scores significantly differed from optimal or 'imposed' prior variance scores (see S2 Table for further details).

Spearman-ranked correlation analysis revealed a significant positive relationship between overall subjective prior variance (across all trials in the main task) and CAPE-P scores in both the discovery ($r = 0.11$, $p = 0.046$, 95%CI = [0.0002, 0.214]) and validation ($r = 0.14$, $p = 1.16 \times 10^{-4}$, 95%CI = [0.069, 0.219]) datasets (Fig 6). This suggests that participants with increasing psychotic-like experiences tend to perceive prior information to be more uncertain or variable across the task. Spearman correlations between CAPE-P scores and subjective prior variance were also calculated across each condition, as shown in Table 3. Whilst significant positive relationships were found across all conditions in the validation dataset, this relationship was only seen in wide likelihood conditions in the discovery dataset.

To consider the impact of working memory on prior precision, we conducted a Spearman-ranked correlation analysis between subjective prior variance and mean spatial span (as a proxy measure of working memory performance). We found a significant, negative correlation between subjective prior variance and mean spatial span the discovery ($r = -0.2$, $p = 2.3 \times 10^{-4}$) and validation dataset ($r = -0.2$, $p = 3.0 \times 10^{-8}$). However, we also found a significant, negative correlation between subjective likelihood variance and mean spatial span in the discovery ($r = -0.17$, $p = 1.4 \times 10^{-3}$) and validation dataset ($r = -0.26$, $p = 5.1 \times 10^{-13}$). This suggests that decreases in working memory might similarly degrade estimates of uncertainty across task in general, rather than prior precision specifically.

**Table 3. Spearman-ranked correlation analyses between subjective prior variance and psychotic-like experiences across each of the four task conditions.**

| CAPE-P score correlated with: | Discovery dataset | | Validation dataset | |
|---|---|---|---|---|
| | $r_s$ | $p$ | $r_s$ | $p$ |
| | Subjective prior variance | | Subjective prior variance | |
| **PnLn** | 0.031 | 0.581 | 0.098 | 0.011 |
| **PnLw** | 0.131 | 0.017 | 0.146 | $7.5 \times 10^{-5}$ |
| **PwLn** | 0.065 | 0.247 | 0.131 | $5.8 \times 10^{-4}$ |
| **PwLw** | 0.123 | 0.027 | 0.125 | $8.4 \times 10^{-4}$ |

## Discussion

This study aimed to investigate the relationship between psychotic-like experiences and aberrant weighting of sensory evidence (likelihood) relative to contextual beliefs (priors) in perceptual decision-making under uncertainty. Orthogonal manipulation of uncertainty in both likelihood and prior information, which is often lacking in previous literature, allowed for a deeper disentanglement and quantification of alterations in the precision-weighting of information, and its association with the non-clinical continuum of psychotic-like experiences. Specifically, we investigated whether an imbalance in the relative precision weighting of likelihood to prior information was driven by greater subjective uncertainty in likelihood information or greater subjective uncertainty in prior information. This extends predictive processing literature, utilised as a fundamental framework for understanding positive symptom formation in schizophrenia, by elucidating whether aberrant learning under uncertainty is also evident across the non-clinical continuum [2,10]. We found a significant, positive relationship between psychotic-like experiences and sensory weighting, indicating an overreliance on likelihood relative to prior information across the task. This relationship was replicated across two large, independent datasets, suggesting robust replicability of our findings. We also found that psychotic-like experiences positively correlated with both subjective likelihood variance *and* subjective prior variance in the larger, validation dataset, insinuating the perception of generally greater task instability. We argue that the heightened reliance on likelihood information throughout the task appears to be driven by noisier representations of prior information (Fig 6), rather than by the sharpening of sensory representations, which were also shown to be noisier as psychotic-like experiences increased (Fig 5). Alternatively, the heightened reliance on likelihood information could lead to the experience of aberrant salience, which could be more generally influencing the perception of greater task instability.

An overreliance on likelihood aligns with the bottom-up account of positive symptom formation in schizophrenia. Early predictive processing accounts suggest that heightened aberrant salience towards objectively uninformative events or stimuli (i.e., likelihood information) results in the misinterpretation of prediction errors as meaningful change [1,2,9]. This misallocation of precision contributes to misleading belief updating and inferences about the environment, resulting in an altered internal model of the world. This is thought to be the foundation of faulty inference, leading to the formation of false concepts as seen in delusions and/or false precepts as seen in hallucinations. Dopaminergic dysfunction encompasses a biologically plausible formulation of this theoretical framework, whereby hyperactivity of phasic dopamine release contributes to aberrant processing of unexpected events or stimuli [28,29]. Interestingly, presynaptic hyperdopaminergic function has not only been found to be related to the severity of symptoms in people with schizophrenia, but also to the degree of schizotypy in healthy individuals [30,31]. Thus, alterations in the processing of predictions errors, closely aligning with an enhanced dopaminergic tone, may contribute to confounding perceptual inferences as seen across the continuum of psychosis. These findings extend the aberrant salience theory of symptom formation into the non-clinical continuum of psychotic-like experiences. The role of dopamine in the precision weighting of uncertainty has been previously investigated with a similar coin task paradigm in patients with Parkinson's disease, which is characterised by low dopaminergic tone [32]. Interestingly, researchers found that dopaminergic medication influenced the weight afforded to sensory information in patients, providing empirical evidence that increasing dopamine levels increases individuals' reliance on likelihood relative to prior information in this population. Future research should verify this assumption across the continuum of psychosis, in order to investigate whether individual differences in sensory weighting correlates with dopaminergic sensitivity and psychotic-like experiences in healthy populations.

Whilst our findings seem to align with a bottom-up account of aberrant sensory processing under uncertainty, we conflictingly found that psychotic-like experiences were associated with increased subjective uncertainty in *both* prior and likelihood information. Firstly, greater subjective prior variance suggests that the heightened reliance on likelihood information might be driven by greater subjective uncertainty in the inferred prior (i.e., decreased precision in prior information). Although this cannot be directly inferred, this aligns with a series of previous studies, demonstrating that a decreased tendency for percept stabilisation whilst viewing ambiguous stimuli was related to the degree of delusional convictions in people with schizophrenia [33], as well as the propensity of delusional ideation in healthy individuals [34]. Our findings also align with previous literature demonstrating a perceptual decision-making bias towards sensory information over priors in patients with schizophrenia [35,36]. Thus, a common attenuation in predictive signalling during perceptual inference substantiates our hypothesis that an overreliance on sensory observations is driven by decreased precision in prior expectations. Despite this, the increased subjective prior variance could also be a result of an overreliance on sensory observations, rather than a driver for it, which cannot be decisively ambiguated from our findings. Moreover, Weilnhammer et al., [37] similarly found that the severity of perceptual anomalies and hallucinations in people with schizophrenia were associated with a shift of perceptual inference towards sensory information and away from prior expectations when deciphering perceptually ambiguous visual stimuli. In fact, deficits in sensory prediction have also been found to correlate with schizotypy and delusion-like thinking in non-clinical samples [38], suggesting that decreased prior precision could be a stable, trait-like characteristic of individuals, rather than a mere consequence of deluded or hallucinatory states in schizophrenia. This not only provides support for the psychosis-continuum perspective in which frank psychosis is considered an extreme outcome of a continuously distributed phenotype [18], but also provides insight into the neurocognitive basis of positive symptom formation in schizophrenia.

Furthermore, although our finding of decreased prior precision could explain the shift in belief updating to favour sensory evidence, this interpretation is complicated by the finding of a similar decrease in likelihood precision. Simultaneous aberrancies of heightened uncertainty in both likelihood and prior information may suggest that people with increasing psychotic-like experiences have general overestimation of uncertainty. In other words, they perceive greater instability or uncertainty in their inferred internal representation of the world [13] (see Fig 7). The misallocation of precision may be hierarchical, such that imprecision in higher order prior beliefs may lead to a lack of regularisation that renders the environment seemingly volatile or unpredictable [2]. This would enhance the weight of lower-level prediction errors, resulting in an overreliance in sensory observations [1,9]. In other words, heightened instability in higher order beliefs may be driving the updating of prediction errors, despite simultaneous instability in the representation of one's sensory environment at lower hierarchical levels. Hence decreased precision in both prior and likelihood information could be explained through the processing of uncertainty at differ layers of the cortical hierarchy. Layer specific neuroimaging techniques would be required for empirical verification, providing a promising avenue for future research [12]. Interestingly, recent research found that high confidence false percepts (a measure of hallucinatory propensity) were related to stimulus-like activity in middle input layers of the visual cortex in healthy participants [39]. Task measures of hallucination propensity were also found to be associated with everyday hallucination severity, suggesting that hallucinatory-like perception may arise from spontaneous bottom-up activity in input layers of the visual cortex. This corroborates neural evidence for the bottom-up account of sensory processing in psychosis.

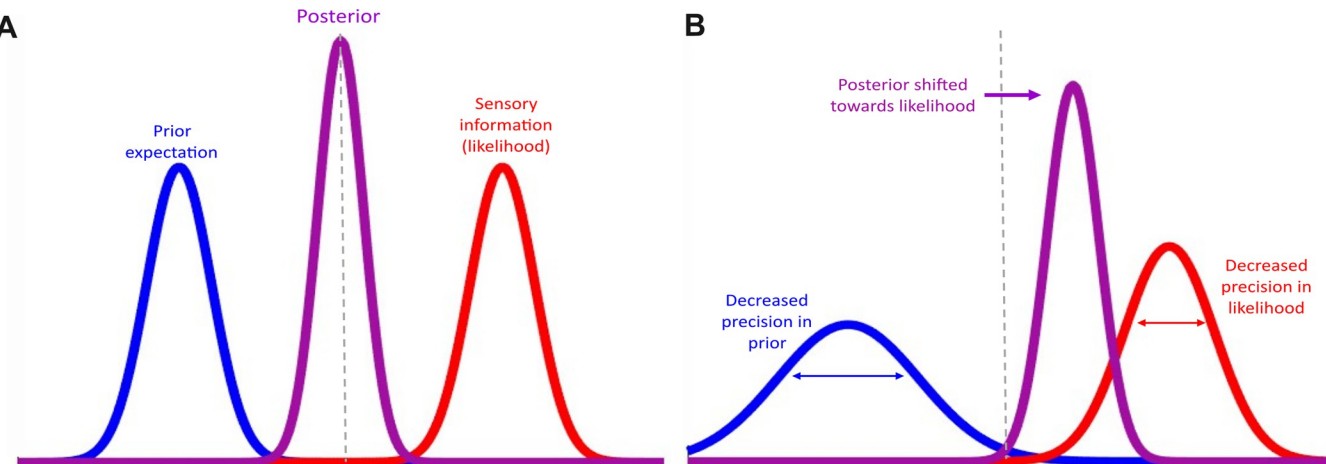

**Fig 7. Visual depiction of the precision weighting of prior and likelihood information found in people with greater psychotic-like experiences.** *A)* Represents an unbiased integration of prior and likelihood information. *B)* Represents proposed explanation of findings, depicting a simultaneous decrease in the precision afforded to prior and likelihood information, where the decrease in prior precision is hypothesized to be driving a shift in the posterior towards likelihood information (i.e., overreliance on likelihood).

Moreover, a wealth of previous literature corroborates our finding of aberrantly perceived task instability across the continuum of psychosis. This is often shown in probabilistic reversal learning paradigms, which demonstrate increased behavioural switching in dynamic or volatile environments as a predictor of subjectively heightened task or environmental instability [15,40]. This has been found in people with schizophrenia [40,41], individuals at risk for psychosis [15], and more recently, was found to correlate with the degree of psychotic-like experiences in a neurotypical sample [42]. Similarly, when characterising belief updating on a trial-by-trial basis, Nassar et al., [43] found that patients with schizophrenia show a generally reduced precision of beliefs and an inflexibility of belief updating. This finding allowed a simultaneous explanation of patients completely ignoring new information and persevering on previous responses (decreased likelihood precision), as well as the overly flexible behavioural adaptation to random noise (decreased prior precision [13]). Thus, our findings align with this study, augmenting evidence that aberrant perception of task instability due to a decreased precision in both prior and likelihood information also extends along the continuum of psychosis into non-clinical populations.

Furthermore, our study found little to no relationship between variability in sensory weighting and psychotic-like experiences from trial-to-trial and across blocks, suggesting that psychotic-like experiences were not necessarily related to aberrant learning or behavioural switching in the utilisation of likelihood to prior information in this task. This is contrary to what we might expect, given the observed association between increasing psychotic-like experiences and a general perception of heightened task instability. To investigate this further, specific manipulations of environmental volatility could provide ecological utility that the current study was somewhat lacking [13,44]. This is because certain links with psychopathology or individual differences may only emerge in an unstable environment [45]. Thus, a more nuanced approach to investigating trial-by-trial variability in sensory weighting should incorporate higher-level of task instability (i.e., volatility), such as blocks where uncertainty in prior information changes more rapidly compared to blocks where this remains stable [43]. Perhaps this level of environmental uncertainty is required to elucidate an association between aberrant switching in sensory weighting and psychotic-like experiences. Additionally, whilst we see a

clear integration of prior and likelihood information in each condition of the task with performance verging towards Bayes optimal, there seems to be a slight overweighting of likelihood information across three of the four conditions (Fig 3). Whilst optimality is not a requirement for Bayesian inference [46], a general bias towards likelihood information is a limit of the task which should be investigated further.

Interestingly, the co-occurrence of enhanced subjective uncertainty in likelihood *and* prior information was only observed in the larger, validation dataset, and not seen in the original discovery dataset. This exemplifies the benefit of conducting an independent replication, demonstrating that a sufficient sample size was critical in capturing the demonstrated results. As psychotic-like experiences are generally sparse in the general population (prevalence of approximately 7%; [47]), it is unsurprising that to capture such small effect sizes, we require a larger sample. Small effect sizes are common in the literature regarding psychotic-like experiences as a trait-like characteristic, with these effects naturally larger in clinical populations. Despite this, the reported subclinical effects are modest but significant and robustly replicated across both datasets. We argue that multiple comparisons such as Bonferroni corrections are too conservative to be applied to this data given that some of the measures are correlated and others determined from independent datasets. Given that four comparisons (at most) are being made between psychotic-like experiences and parameters of interest across the analyses, adjusted p-values with Bonferroni corrections can be seen in S3. This shows that the positive relationship between psychotic-like experiences and subjective likelihood variance, as well as subjective prior variance, remain significant in the larger, validation dataset with these corrections applied. Instead of this approach, we have incorporating bootstrapped confidence intervals (with 1000 bootstrapped replicates) into our analyses (also shown in S3 Table), which we use for our interpretations.

Additionally, whilst our findings support the notion that aberrant sensory predictions extend into the non-clinical continuum of psychosis, paradigms such as the coin task that orthogonally manipulate the uncertainty of likelihood and prior information have not yet been empirically tested in clinical samples. Thus, a limit of the current study is that, whilst it is advantageous to investigate continuously distributed phenomena in healthy populations, it does not consider distinct neurocognitive discontinuities that may exist between subclinical and clinical populations [48]. A cross-sectional design sampling from patients with schizophrenia, first episode psychosis, and clinical high risk for psychosis could potentially differentiate the precision afforded to uncertainty across disease trajectory. Therefore, empirical verification is required to explicitly test the relative weighting of likelihood and prior information with the emergence and formation of psychosis across the entire continuum of psychosis. Another limitation of the study is the lack of attention checks throughout the paradigm. Despite this, we included quality control measures for online testing such as removing participants with particularly low mean estimation error scores, as this was deemed to be an indicator of poor adherence or engagement with the task. Additionally, participants were asked to complete practice tasks prior to completion of the main task, to ensure that they understood what was required in the main task.

Additionally, competing hypotheses have provided contradictory perspectives on the emergence of hallucinations compared to delusions in schizophrenia [2]. Whilst delusion formation is often characterised as an overreliance on sensory observations leading to the formation of false concepts [49], hallucination formation has contrastingly been characterised as abnormally strong priors, leading to the formation of false percepts [10]. Although our data is not consistent with the overweighting of priors account, our participants did not experience actual hallucinations as they were sampled from a non-clinical population, hence we cannot completely rule out this account in hallucinating individuals. Additionally, heterogeneity in individuals' proneness to hallucination-like percepts compared to delusional-like ideation was

not considered in our neurotypical sample. As this may influence one's relative reliance on priors to likelihoods across the task, this could provide an alternative explanation for the simultaneous decrease in prior and likelihood precision found in our study and could also account for heterogenous findings in previous literature [50]. In fact, one study showed that the use of prior knowledge varies with the composition of psychotic-like phenomena (in terms of aberrant percepts vs aberrant beliefs) in healthy individuals [51]. Contrary to this, a recent study found evidence for a reduced reliance on priors to sensory evidence in relation to both delusion *and* hallucination proneness [52], which is consistent with our data and proposed model for the continuum of psychosis (Fig 7). We followed up these findings in the current study to determine whether there was a similar relationship in CAPE-P subscale scores, namely bizarre experiences, delusional ideation, and perceptual anomalies [53] (see S2 Fig for a distribution of the subscales in our data). We found that delusional ideation was positively correlated with subjective prior variance in both discovery and validation sets (see S4 Table and S5 Table). We also found other significant, yet unspecific, relationships although only for the validation set. However, one should be cautious when interpreting these findings since these subscales are based on limited number of items (bizarre experiences are defined by 7 items, delusional ideation is defined by 9 items, and perceptual anomalies are defined by 4 items [53]). Hence, we turn to the interpretation of the 20 item CAPE-P instead and suggest future research to employ separate measures to compare hallucinatory and delusional-like experiences, and their respective association with the precision weighting of likelihood to prior information in the coin task. Examples of specific measures include Peters Delusion Inventory [54] and Cardiff Anomalous Perceptions Scale [55] which are designed to measure delusional ideation and hallucinatory experiences in the general population respectively.

In conclusion, our findings provide evidence that psychotic-like experiences are associated with an overweighting of sensory evidence relative to prior expectations, which seem to be driven by decreased precision in prior information. Our findings suggest that psychotic-like experiences are associated with aberrant precision of encoding uncertainty in both prior and likelihood information. This provides an interesting platform for understanding and quantifying aberrancies in perceptual processing under uncertainty, and how this relates to the nonclinical continuum of psychosis.

## Supporting information

**S1 Fig. Comparison of mean estimation error performance across trial conditions in A) the discovery and B) the validation datasets.** Demonstrating the variability in mean estimation errors across each of the four the trial conditions to supplement task performance analyses. Conditions: PnLn = narrow prior, narrow likelihood (red dots); PnLw = narrow prior, wide likelihood (green dots); PwLn = wide prior, narrow likelihood (teal dots); PwLw = wide prior, wide likelihood (purple dots).
(TIFF)

**S2 Fig.** Distribution of CAPE-P subscale scores (bizarre experiences, delusional ideation and perceptual anomalies) in A) the discovery dataset (n = 363) and B) validation dataset (n = 782).
(TIFF)

**S1 Table. Median sensory weights significantly different from Bayesian optimal sensory weights across all conditions in both discovery and validation datasets.**
(PDF)

**S2 Table. Median subjective prior variance scores significantly different from Bayesian optimal or 'imposed' prior variance scores across all conditions in both discovery and**

**validation datasets.**
(PDF)

**S3 Table. Adjusted p-values with Bonferroni corrections for multiple comparisons applied to main correlations of interest.** In the manuscript, correlation analyses compare the associations between psychotic-like experiences with (1) sensory weight, (2) variability in sensory weight, (3) subjective likelihood variance (although this is calculated from a separate task) and (4) subjective prior variance. Given this, four comparisons are (at most) being made across the analyses. Adjusted p-values for these comparisons are demonstrated here, along with bootstrapped confidence intervals with 1000 bootstrapped replicates.
(PDF)

**S4 Table. Spearman correlation between CAPE-P subscale scores (bizarre experiences, delusional ideation and perceptual anomalies) and variables of interest (sensory weight, subjective likelihood variance and subjective prior variance) in the discovery dataset (n = 363).**
(PDF)

**S5 Table. Spearman correlation between CAPE-P subscale scores (bizarre experiences, delusional ideation and perceptual anomalies) and variables of interest (sensory weight, subjective likelihood variance and subjective prior variance) in the validation dataset (n = 782).**
(PDF)

## Author Contributions

**Conceptualization:** Isabella Goodwin, Joshua Kugel, Robert Hester, Marta I. Garrido.

**Data curation:** Isabella Goodwin.

**Formal analysis:** Isabella Goodwin.

**Funding acquisition:** Marta I. Garrido.

**Investigation:** Isabella Goodwin, Joshua Kugel.

**Methodology:** Isabella Goodwin, Joshua Kugel, Marta I. Garrido.

**Project administration:** Robert Hester, Marta I. Garrido.

**Resources:** Marta I. Garrido.

**Software:** Isabella Goodwin.

**Supervision:** Robert Hester, Marta I. Garrido.

**Validation:** Isabella Goodwin.

**Visualization:** Isabella Goodwin.

**Writing – original draft:** Isabella Goodwin.

**Writing – review & editing:** Robert Hester, Marta I. Garrido.

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
