## [Decision Letter · Decision Letter 0]

18 Jun 2023

Dear Ms Goodwin,

Thank you very much for submitting your manuscript "Bayesian accounts of perceptual decisions in the nonclinical continuum of psychosis: Greater imprecision in both top-down and bottom-up processes" for consideration at PLOS Computational Biology.

As with all papers reviewed by the journal, your manuscript was reviewed by members of the editorial board and by several independent reviewers. In light of the reviews (below this email), we would like to invite the resubmission of a significantly-revised version that takes into account the reviewers' comments.

We cannot make any decision about publication until we have seen the revised manuscript and your response to the reviewers' comments. Your revised manuscript is also likely to be sent to reviewers for further evaluation.

Sincerely,

Christoph Mathys

Academic Editor

PLOS Computational Biology

Lyle Graham

Section Editor

PLOS Computational Biology

Reviewer's Responses to Questions

**Comments to the Authors:**

Reviewer #1: General comment

In this study the authors used an online psychophsyical experiment to investigate the neurocomputational mechanisms underlying psychosis proneness in two large samples of healthy participants, employing a task designed to test prior and likelihood estimates independently. The main result, which is found consistently in both samples, is an increased weighting of the likelihood relative to prior information in relation to psychosis-proneness. In addition, psychosis-proneness is reported to be associated with reduced perceived task stability as reflected by both subjective likelihood variance and subjective prior variance. The authors interpret their findings as evidence for both weaker bottom-up and top-down perceptual processes that leads to an overweighting of sensory evidence relative to prior information.

The study is timely and tackles an important question in the context of predictive processing accounts of psychosis. The task design and experimental methods are sound and the use of independent discovery and confirmation samples lends particular strength to the findings. So overall, there is much to like about this study. The only major concerns I have relate to how the measures of prior and likelihood precision are derived and whether the conclusions that are based on these measures are tenable. As it stands, I’m therefore not sure whether the novel part of the findings, relating to prior precision, is supported by the data.

Specific points

1. I have to admit that I struggled to understand the rationale for (and computation of) subjective task variance. This is a critical measure, because much of the author’s interpretation of the findings is based on it. But how can subjective prior variance be derived from actual likelihood variance and optimal sensory weights? This really needs to be unpacked, so the reader can understand better what the measure of subjective prior variance really means and how it should be interpreted.

2. Relatedly, the authors conclude that “that the perceived instability in prior information is driving a greater reliance on likelihood information throughout the task”. This interpretation is, in my opinion, not tenable. How do we know that this is the direction of causality? Is it indeed perceived instability of priors that drives overreliance? Or may it be the other way around, i.e., could a primary overreliance on sensory information (e.g. due to a heightened dopamine availability, as discussed by the authors) lead to the experience of aberrant salience and consequently to reduced subjective task stability? Whether the latter scenario is a likely one or not, the causal conclusion that subjective prior variance drives overreliance on sensory evidence does not seem to be supported by the data.

3. Again in relation to the measures of perceived variance: It is laudable that the task modulates uncertainties in likelihood and prior information independently, but while there is clear evidence for an overreliance on sensory information, no direct measure is available for the reliance on prior information. Instead, a measure of subjective variance is used. It is unclear however, whether this subjective task instability indeed reflects reduced prior precision. To my mind, this is only one possible interpretation. Couldn’t increased subjective prior variance also be a result of overreliance on sensory information (see above)? Wouldn’t it be conceivable that the prior variance is perceived as high, but that this does not directly affect the estimation of prior precision in each trial, i.e. in the actual inferences made? I would like the authors to provide a more balanced discussion of this issue and to tone down the interpretation of increased subjective prior variance as directly reflecting reduced prior precision.

4. I found parts of the introduction a bit difficult to follow:

“…discrepancies within the literature describe both increases and decreases in the relative weighting of likelihood to prior information” (lines 94 ff.). These discrepancies do not follow from what is said in the two sentences before, but rather from what is said in the preceding paragraph.

“…to determine whether an overreliance on prior or likelihood information is driven by more precise priors or more precise likelihoods” (lines 116 f.). This is a muddled sentence that needs to be unpacked. How can “prior or likelihood information” be driven by “precise priors of ore precise likelihood”?

5. Task instructions: Did the participants know, as in were explicitly told, which block they were in? Was the order of blocks randomized or alternating (aiding the participant’s knowledge regarding block structure). Or did they have to infer the thrower (and hence the prior) from the distribution of throws?

6. Regarding the authors’ suggestion for future studies to look at separate measures for delusion and hallucination proneness (lines 575 ff.): They may refer to a recently published study that did this and found evidence for a reduced reliance on priors relative to sensory evidence in relation to both delusion and hallucination proneness, using perceptual decision making tasks (https://pubmed.ncbi.nlm.nih.gov/36440751/).

Reviewer #2: I read and enjoyed Goodwin and colleagues report of their experiment on the use of likelihoods and priors in perceptual decision-making in the general population and how they relate to psychosis-like experiences

They report that sensory inputs are overweighted in a manner that correlates with CAPE-P scores.

This is a potentially useful contribution to a growing empirical literature

However, I have some substantive to concerns which I would like the reviewers to address with new analyses .

First, there is a clear multiple comparisons problem, and the effects they report are quite small - when appropriate correction for multiple correlations is applied, I suspect many (at least in the discovery sample) would not survive.

More importantly, I think there is some misunderstanding about the different kinds of Prior overweighting theories. There are some that suggest strong priors would induce aberrant prediction errors - which then relate to delusions (though this is not a commonly espoused theory). Instead, prior overweighting (due to under-precise inputs or over-precise priors) is more commonly associated with hallucinations. Did any of these participants actually have hallucinations? In the absence of hallcuinations, we might not expect prior overweighting.

One way to address these issues with the data in hand: it would be helpful to know what the paranoia, bizarre ideas, and perceptual aberrations subscake scores were, and whether, people with more perceptual aberrations overweighted, and whether the paranoid participants had stringer volatility beliefs (as has been shown repeatedly by other labs).

Perhaps the authors could pre-register some predictions about how specific symptoms might relate to behavior in their task before analyzing the subscale scores?

**Have the authors made all data and (if applicable) computational code underlying the findings in their manuscript fully available?**

Reviewer #1: None

Reviewer #2: Yes

PLOS authors have the option to publish the peer review history of their article (what does this mean?). If published, this will include your full peer review and any attached files.

Reviewer #1: No

Reviewer #2: No
---

## [Decision Letter · Decision Letter 1]

3 Oct 2023

Dear Ms Goodwin,

Thank you very much for submitting your manuscript "Bayesian accounts of perceptual decisions in the nonclinical continuum of psychosis: Greater imprecision in both top-down and bottom-up processes" for consideration at PLOS Computational Biology. As with all papers reviewed by the journal, your manuscript was reviewed by members of the editorial board and by several independent reviewers. The reviewers appreciated the attention to an important topic. Based on the reviews, we are likely to accept this manuscript for publication, providing that you modify the manuscript according to the review recommendations.

Sincerely,

Christoph Mathys

Academic Editor

PLOS Computational Biology

Lyle Graham

Section Editor

PLOS Computational Biology

Reviewer's Responses to Questions

**Comments to the Authors:**

Reviewer #1: I would like to thank the authors for fully addressing all my concerns. I am looking forward to seeing this excellent paper published.

Reviewer #3: In this study, across two population datasets, psychotic-like experiences (positive symptoms in the CAPE) were associated with

- >weighting of sensory information

- loss of subjective prior precision

- loss of subjective likelihood precision (in the validation study only)

- (no relation to 'prior sensitivity', i.e. changes in sensory weight from block to block as prior precision changes)

I found this paper interesting and a useful contribution, given it is rare that people design experiments to manipulate prior and likelihood precision at the same time in the schizophrenia spectrum, yet very important to do so. I thought the authors' responses to the other reviewers concerns were valid. The use of two large samples is reassuring: the effects are quite small but the replication sample assuages any multiple comparison concerns that I might have had.

I don't have suggestions for major revisions but there are a few points that I would like to be addressed:

p15 - I am perhaps misunderstanding something but why does the likelihood-only task show large effects of likelihood precision on estimation errors, but the main task doesn't show these effects or effects of prior precision (except in the validation dataset)?

p16 - the boxes in the plots in Fig 3 are not visible, I suggest they are turned to black, or a much darker shade?

p19 - Why is a Spearman correlation used in Fig 5 in the validation set but not in the training set? Also it would be helpful to plot each of the various training/validation data figures on the same axes to facilitate comparison.

Discussion - the move from decreased subjective precision of likelihood and prior to the concept of 'task instability' is quite a big one, not really supported by evidence. As the authors say, their task is not really set up to assess subjective 'task instability' (defined as greater subjective volatility of contingencies) and in any case they don't see evidence for this in the data they have. So I don't think the authors should state in the abstract and final conclusion etc that they have found that psychotic-like experiences are associated with "greater perceived task instability".

Was there any relationship between subjective prior precision and working memory performance? Does the latter explain all the group differences in the former? One might expect that failure to remember previous trials would degrade prior precision but not likelihood precision... Also, were there any attention checks in the paradigm? If some more schizotypal participants just weren't paying much attention to the task, presumably their results would look like the obtained ones? If there weren't attention checks then it should be mentioned as a limitation.

Last, the authors don't cite some earlier attempts to devise tasks that distinguished between prior and likelihood precision in psychiatric samples: Karvelis et al (2018, eLife) - who found no differences in subjective prior/likelihood precision in schizotypal traits (but did in autistic traits), and also Valton et al (2019, Brain) who found that Scz patients seemed to learn (visual) priors of equal precision to controls but these priors had less influence (vs those of controls) when the sensory info was weak. They could also cite Jardri et al (2017 Nat Hum Beh) who also found evidence of sensory bias over priors, and also a relation of working memory to prior weighting (but not increased likelihood variance).

**Have the authors made all data and (if applicable) computational code underlying the findings in their manuscript fully available?**

Reviewer #1: Yes

Reviewer #3: Yes

PLOS authors have the option to publish the peer review history of their article (what does this mean?). If published, this will include your full peer review and any attached files.

Reviewer #1: No

Reviewer #3: No

Figure Files:

Data Requirements:

Reproducibility:

References:

---

## [Decision Letter · Decision Letter 2]

7 Nov 2023

Dear Ms Goodwin,

We are pleased to inform you that your manuscript 'Bayesian accounts of perceptual decisions in the nonclinical continuum of psychosis: Greater imprecision in both top-down and bottom-up processes' has been provisionally accepted for publication in PLOS Computational Biology.

Best regards,

Christoph Mathys

Academic Editor

PLOS Computational Biology

Lyle Graham

Section Editor

PLOS Computational Biology

Reviewer's Responses to Questions

**Comments to the Authors:**

Reviewer #1: No further comments.

Reviewer #3: Thanks - I have no further comments.

**Have the authors made all data and (if applicable) computational code underlying the findings in their manuscript fully available?**

Reviewer #1: None

Reviewer #3: Yes

PLOS authors have the option to publish the peer review history of their article (what does this mean?). If published, this will include your full peer review and any attached files.

Reviewer #1: No

Reviewer #3: No

---

## [Editor Report · Acceptance letter]

15 Nov 2023

PCOMPBIOL-D-23-00202R2 

Bayesian accounts of perceptual decisions in the nonclinical continuum of psychosis: Greater imprecision in both top-down and bottom-up processes

Dear Dr Goodwin,

I am pleased to inform you that your manuscript has been formally accepted for publication in PLOS Computational Biology. Your manuscript is now with our production department and you will be notified of the publication date in due course.

With kind regards,

Lilla Horvath
